# Novel Sol-Gel Route to Prepare Eu^3+^-Doped 80SiO_2_-20NaGdF_4_ Oxyfluoride Glass-Ceramic for Photonic Device Applications

**DOI:** 10.3390/nano13050940

**Published:** 2023-03-05

**Authors:** María Eugenia Cruz, Thi Ngoc Lam Tran, Alessandro Chiasera, Alicia Durán, Joaquín Fernandez, Rolindes Balda, Yolanda Castro

**Affiliations:** 1Instituto de Cerámica y Vidrio, CSIC, 28049, Madrid, Spain; 2IFN-CNR CSMFO Laboratory and FBK Photonics Unit, Via alla Cascata 56/C Povo, 38123 Trento, Italy; 3Department of Physics, Politecnico di Milano, Piazza Leonardo da Vinci 32, 20133 Milano, Italy; 4Department of Materials Technology, Faculty of Applied Science, Ho Chi Minh City University of Technology and Education, Vo Van Ngan Street 1, Thu Duc District, 720214 Ho Chi Minh City, Vietnam; 5Donostia International Physics Center (DIPC), 20018 San Sebastian, Spain; 6Department Física Aplicada, Escuela Superior de Ingeniería, Universidad del País Vasco (UPV-EHU), 48013 Bilbao, Spain; 7Centro de Física de Materiales, (CSIC-UPV/EHU), 20018 San Sebastian, Spain

**Keywords:** oxyfluoride glass-ceramics OxGCs, sol-gel pre-crystallised nanoparticles route, NaGdF_4_

## Abstract

Oxyfluoride glass-ceramics (OxGCs) with the molar composition 80SiO_2_-20(1.5Eu^3+^: NaGdF_4_) were prepared with sol-gel following the “pre-crystallised nanoparticles route” with promising optical results. The preparation of 1.5 mol % Eu^3+^-doped NaGdF_4_ nanoparticles, named 1.5Eu^3+^: NaGdF_4_, was optimised and characterised using XRD, FTIR and HRTEM. The structural characterisation of 80SiO_2_-20(1.5Eu^3+^: NaGdF_4_) OxGCs prepared from these nanoparticles’ suspension was performed by XRD and FTIR revealing the presence of hexagonal and orthorhombic NaGdF_4_ crystalline phases. The optical properties of both nanoparticles’ phases and the related OxGCs were studied by measuring the emission and excitation spectra together with the lifetimes of the ^5^D_0_ state. The emission spectra obtained by exciting the Eu^3+^-O^2−^ charge transfer band showed similar features in both cases corresponding the higher emission intensity to the ^5^D_0_→^7^F_2_ transition that indicates a non-centrosymmetric site for Eu^3+^ ions. Moreover, time-resolved fluorescence line-narrowed emission spectra were performed at a low temperature in OxGCs to obtain information about the site symmetry of Eu^3+^ in this matrix. The results show that this processing method is promising for preparing transparent OxGCs coatings for photonic applications.

## 1. Introduction

The development of transparent glass-ceramics (GC) has drawn the attention of numerous researchers, and they have been extensively studied due to their important optical applications [1]. GCs can be used for light-emitting diodes, solar cells, sensing catalysis or in biomedical materials [2,3]. Particularly, when fluoride nanocrystals smaller than 40 nm are dispersed in an oxide glass matrix, the resulting materials are transparent GCs, called oxyfluoride glass-ceramics (OxGCs), which have attractive properties [4]. These materials are transparent from the visible to near-infrared region and are compatible with new optical devices for communications or sensors. Furthermore, OxGCs retain the most relevant properties of glasses—good mechanical and thermal stability—and present properties characteristic of fluoride crystals as an effective optical media for light propagation and luminescence enhancement, and a high linear and nonlinear refractive index [5,6,7,8]. The crystal structure of fluorides as hosts of RE ions reduces the probabilities of multiphonon relaxation, resulting in high luminescence efficiencies [9,10,11,12]. Thus, the efficiency of the emission, optical transmission and spectra profile can be modified depending on the fluoride phase and the dopant. Among fluoride nanoparticles, those with the formula ALnF_4_ (where A is an alkaline element and Ln a lanthanide) have become more attractive in comparison to other fluorides due to their lowest phonon energies and wide band gap (9–10 eV) [13]. Particularly, NaGdF_4_ nanoparticles have been prepared both in cubic (α-phase) and hexagonal (β-phase). Thoma, B.R.E. et al. [14] reported the influence of a precursors ratio to prepare either a cubic or hexagonal phase, and later, You, F. et al. [15] presented the lattice parameters of both phases and studied the optical behaviour of the Eu^3+^-doped nanoparticles with promising results. In the last decade, many authors reported the preparation of RE-doped NaGdF_4_ nanoparticles and their luminescent properties, showing their excellent behaviour for both down and up conversion processes [16,17,18]. Oxyfluoride glass-ceramics with NaGdF_4_ have also been prepared using a melting-quenching process (MQ) [19]. The preparation of GCs via MQ requires the melting of inorganic raw materials at high temperatures (1400 °C–1700 °C) followed by a controlled heat treatment for generating precipitation and crystal growth. In 2013, Herrmann, A. et al. [20] obtained the NaGdF_4_ in cubic and **hexagonal** phases by heat treatment at 750 °C. One of the limitations of the MQ process is the low maximum amount of active phase that can be obtained due to the volatilisation of fluorides at the high temperatures needed in the process. In addition, in many cases, MQ materials require very long heat treatments (3 h–80 h) to reach the desired crystallisation. Moreover, OxGCs prepared by MQ are usually bulk materials, and this processing method is not suitable for preparing coatings [20,21,22,23]. 

Sol-gel (SG) appeared as an alternative method, emphasised in the chemistry of the process, to avoid the drawbacks of MQ [24,25,26]. It is a favourable process through which highly homogeneous materials can be obtained at low temperatures (<500 °C) [27,28]. Furthermore, various material forms, e.g., bulks, powders and coatings, can be processed through the hydrolysis and polycondensation of metal alkoxide precursors, such as tetraethyl orthosilicate (TEOS) in a solvent, typically alcohol. Sol-gel allows us to control the structures in the molecular scale of the materials throughout the whole process, allowing for the preparation of highly homogeneous materials [29].

In 2019, Velázquez, J.J. et al. [13] reported the preparation of novel transparent OxGCs with NaGdF_4_ using the sol-gel technique. The authors reported the precipitation of β-NaGdF_4_ crystals through the heat treatment of a sol prepared in two steps, commonly called the TFA route [30]. In this case, silica precursors are mixed with a solution of rare-earth precursors in TFA, followed by a controlled heat treatment. The optical results obtained showed an efficient energy transfer from Gd^3+^ to Eu^3+^ as well as comparable lifetime values compared with the literature. Nevertheless, the OxGCs materials prepared using the TFA route are limited to transparent self-supported layers and powders. Although transparent and homogeneous OxGCs coatings were prepared using the TFA route, the densification of the films occurred during the heat treatment and concurrently with the crystallisation process, resulting in crystals that were too small, with sizes below 3 nm. The emission spectra of these OxGCs coatings revealed that rare-earth ions were located in an amorphous environment, suggesting that they were not successfully incorporated into the crystals, probably due to their small size [31]. 

In 2020, the GlaSS group from the Instituto de Cerámica y Vidrio (CSIC) reported an alternative route, labelled as the “pre-crystallised nanoparticles route” [30], based on the previous synthesis of crystalline fluoride nanoparticle aqueous suspensions that are subsequently incorporated into a silica sol [32]. Using this method, it is possible to control the morphology and size of the nanocrystals before their incorporation into the silica matrix. In a previous work, Cruz et al. [32] reported the preparation of Nd^3+^-doped 80SiO_2_-20LaF_3_ oxyfluoride glass-ceramic powders via the pre-crystallised nanoparticles route with encouraging photonic results. The authors studied the stability of the LaF_3_ nanoparticles in the silica matrix before and after the heat treatment at 450 °C, reporting the best luminescence results after the heat treatment when organics are eliminated. The optical characterisation of the OxGCs powders was also reported, with similar lifetimes to those for nanoparticles. Later, the GlaSS group reported the preparation of OxGCs coatings with the composition Nd^3+^-doped 80SiO_2_-20LaF_3_, obtaining a bulk-like lifetime value of 440 µs [33].

A key point of the “pre-crystallised nanoparticles route” is the processing of the fluoride nanoparticle suspensions. Some authors reported the preparation of NaGdF_4_ nanoparticle suspensions [34,35,36,37]. In 2004, Mech et al. [38] stated for the first time the preparation via hydrothermal synthesis of Eu^3+^-doped NaGdF_4_ and KGdF_4_ nanoparticles [38,39]. They prepared the nanoparticles by mixing the rare-earth oxides in a hydrochloric acid solution with further incorporation of sodium fluoride. The results were auspicious, showing good optical properties, with sharp and well-defined peaks for Eu^3+^ emissions through the excitation of Gd^3+^. Later, Sudheendra, L. et al. [40] described the preparation of Eu^3+^-doped NaGdF_4_ using a sodium citrate solution, indicating that the Gd^3+^-Eu^3+^ host–dopant couple is the best system for down-conversion after UV excitation as the emission energy transition within Gd^3+^ can resonantly couple to the excited state of Eu^3+^ ions. The authors demonstrated that in Eu^3+^-doped NaGdF_4_ nanoparticles, the emission of the hexagonal phase is 25% more intense than those of the cubic structure. Although different papers reported the preparation of powdered NaGdF_4_ nanoparticles via hydrothermal synthesis in an autoclave controlling the pressure and temperature [40,41,42], only a few papers have reported the use of this process to obtain aqueous nanoparticle suspensions to control the morphology of the nanoparticles.

The objective of this work was the preparation and characterisation of optically active oxyfluoride glass-ceramics through the incorporation of Eu^3+^-doped NaGdF_4_ nanoparticle aqueous suspensions into a silica sol. In this work, powdered glass-ceramics were prepared, and the suitability of this processing method to prepare transparent OxGCs was demonstrated. This is a promising method for preparing transparent coatings and overcoming the drawbacks of the traditional melting-quenching method. Moreover, the selection of Eu^3+^ ions as a dopant is due to its excellent properties as a local probe, providing important information on the environmental structure where it is located [43]. 

## 2. Materials and Methods

### 2.1. Synthesis of 1.5Eu^3+^: NaGdF_4_ Aqueous Suspensions

Nanoparticle suspensions of NaGdF_4_ undoped and doped with 1.5 mol. % Eu^3+^ were prepared by mixing gadolinium nitrate (Gd (NO_3_)_3_, Merk) with europium acetate (Eu(CH_3_CO_2_)_3_, Merk) in a molar ratio of 1Gd(NO_3_)_3_: 0.015Eu(CH_3_CO_2_)_3_ with 20 mL of water at room temperature to obtain a homogeneous solution. After 30 min of stirring, 1 g of sodium fluoride (NaF, Merk) was incorporated into the solution, before stirring for another 15 min at room temperature. The final solution was transferred to a Teflon cup and put into a stainless-steel autoclave, heated up to 180 °C and maintained for 20 h and 24 h. The suspensions were collected and used directly. The nanoparticles were labelled according to the reaction time: 1.5Eu^3+^: NaGdF_4_-20 or 1.5Eu^3+^: NaGdF_4_-24.

### 2.2. Synthesis of OxGCs with Composition 80SiO_2_-20(1.5Eu^3+^: NaGdF_4_)

OxGCs with the composition 80SiO_2_-20(1.5 Eu^3+^: NaGdF_4_) were prepared following the “pre-crystallised nanoparticles route” [30]. First, tetraethyl orthosilicate (TEOS, Sigma Aldrich) and methyl-triethoxysilane (MTES, ABCR) were mixed with a molar ratio of 1TEOS:1MTES, followed by the incorporation of the previously prepared aqueous nanoparticle suspensions, 1.5Eu^3+^: NaGdF_4_-24, to reach a final molar relation of 80SiO_2_-20(1.5 Eu^3+^: NaGdF_4_). After the incorporation of the nanoparticle suspensions, concentrated hydrochloric acid (HCl, Sigma Aldrich) was added under vigorous stirring to catalyse the hydrolysis and condensation reactions. The solution was immersed in an ice bath for 2 min to stop the reaction. After that, the sol was stirred for 15 min at room temperature. Finally, absolute ethanol was used to dilute the final sol up to a final concentration of 171 g L^−1^.

### 2.3. Characterisation of 1.5Eu^3+^: NaGdF_4_ Aqueous Suspensions

1.5Eu^3+^: NaGdF_4_-20 and 1.5Eu^3+^: NaGdF_4_-24 suspensions were centrifuged at 6000 rpm for 5 min, and the resulting powders were rinsed with deionised water; the rinsing process was repeated three times. The powders were dried at 75 °C overnight, then heat-treated at 450 °C and 550 °C for 5 h in an oven in an air atmosphere and further characterised.

X-ray diffractions were used to characterise the powders of both compositions Eu^3+^: NaGdF_4_-20 heat-treated at 450 °C for 5 h and Eu^3+^: NaGdF_4_-24 heat-treated at 450 °C and 550 °C for 5 h using an X-ray powder diffractometer (D8 advance, Bruker) with CuK_α_ radiation (λ = 1.5406 A). The diffraction patterns were acquired in the range 10° < 2Ɵ < 70° with a step of 0.03°. The nano-crystalline sizes were calculated by using the Scherrer equation shown in Equation (1).
(1)$$D_{\mathrm{hkl}}=\frac{k\lambda }{\sqrt{\beta^{2}-b^{2}}-\cos\theta }$$
where D_hkl_ is the calculated crystallite size, k = 0.94 for spherical crystals, θ is the Bragg angle, β is the full width of the diffraction peak at half maximum intensity (FWHM) and b is the correction of the instrument. The fits were performed using Origin software and the pseudo-Voigt function.

The size was also calculated using the Williamson–Hall (W-H) plot from which the strain broadening was estimated. To perform the W-H plot, the peak width was studied as a function of 2θ degree. The strain (ℰ) was calculated using Equation (2) for each peak of the XRD pattern. By using the calculated ℰ and the D_hkl_ taken from the Scherrer equation (Equation (1)) for all peaks, a linear regression was generated with its corresponding slope and y-intercept. Considering that the linear regression corresponds to Equation (3), the strain component was calculated from the slope and the crystallite size from the y-intercept [33,34,44].
(2)$$E=\frac{\beta }{4\cos\theta }$$
(3)$$\beta \cos\theta =K\lambda \frac{1}{D_{\mathrm{hkl}}}+4E\sin\theta$$

Fourier transform infrared spectroscopy (FTIR) spectra of undoped NaGdF_4_ NPs, 1.5Eu^3+^: NaGdF_4_-24 dried at 75 °C overnight and 1.5Eu^3+^: NaGdF_4_-24 heat-treated at 450 °C for 5 h were recorded using Perkin–Elmer spectrum 100 FT-IR equipment, in the range of 4000 cm^−1^–450 cm^−1^ with a resolution of 4 cm^−1^.

High-Resolution Transmission Electron Microscopy (HRTEM) was used to characterise 1.5Eu^3+^: NaGdF_4_ NPs heat-treated at 450 °C for 5 h. The powder was re-dispersed in ethanol and then dripped onto a carbon-coated copper grid (Lacey Carbon, LC-200-Cu 25/pk). HRTEM images were taken with a HRTEM-JEO 2100 microscope, and the particle size distribution was determined through ImageJ^®^ software, using a maximum of 10 images. The lattice parameters of NaGdF_4_ were determined using the Image-J source.

### 2.4. Characterisation of OxGCs with Composition 80SiO_2_-20(1.5Eu^3+^: NaGdF_4_)

The OxGCs with the composition 80SiO_2_-20(1.5 Eu^3+^: NaGdF_4_) was dried at 75 °C overnight and then heat-treated at 450 °C for 5 h.

Structural properties of OxGCs powders were studied using X-ray diffraction (XRD) following the procedure described in Section 2.3. In addition, FTIR spectra of 80SiO_2_-20(1.5Eu^3+^: NaGdF_4_) OxGCs before and after the heat treatment were also performed following the setup described in Section 2.3.

### 2.5. Optical Characterisation

Room temperature emission and excitation spectra, as well as luminescence decays of 1.5Eu^3+^: NaGdF_4_ NPs and OxGCs with the composition 80SiO_2_-20(1.5Eu^3+^: NaGdF_4_), with both powders heat-treated at 450 °C for 5 h, were recorded by using a FS5 fluorescence spectrometer (Edinburg Instruments Ltd., UK) equipped with a 150 W xenon lamp. The lifetime value estimation was performed by fitting the exponential decay. The emission was detected using a Hamamatsu R928P photomultiplier.

The absolute photoluminescence quantum yield (PL QY) measurements were studied by using Hamamatsu Quantaurus-QY C11347-11, utilising an integrated sphere to measure all luminous flux. The excitation source was a 150 W xenon light source. The absolute photoluminescence quantum yield was measured under excitation wavelength λexc = 270 ± 10 nm, and the considered PL range for the QY calculation was from 400 nm to 700 nm. The procedure of each measurement was as follows: (i) measurement of a quartz Petri dish with cap, (ii) measurement of the quartz Petri disk containing the powder with cap, and (iii) calculation of the PL QY, following Equation (4).
(4)$$\mathrm{PL}\;\mathrm{QY}=\frac{\mathrm{Number}\;\mathrm{of}\;\mathrm{photons}\;\mathrm{emitted}\;\mathrm{as}\;\mathrm{PL}\;\mathrm{from}\;\mathrm{the}\;\mathrm{sample}}{\mathrm{Number}\;\mathrm{of}\;\mathrm{photons}\;\mathrm{absorbed}\;\mathrm{by}\;\mathrm{the}\;\mathrm{sample}}$$

This measurement was repeated 10 times, and the final value of absolute photoluminescence quantum yield was obtained as the average of the 10 measurements.

Resonant time-resolved fluorescence line-narrowed (TRFLN) spectra of the ^5^D_0_→^7^F_0,1,2_ transitions of Eu^3+^ were performed by exciting the OxGCs with the composition 80SiO_2_-20(1.5Eu^3+^: NaGdF_4_) heat-treated at 450 °C for 5 h into the ^7^F_0_→^5^D_0_ transition with a pulsed frequency doubled Nd: YAG pumped tuneable dye laser of 9 ns pulsed width and 0.08 cm^−1^ linewidth and detected by an EGG&PAR Optical Multichannel Analyzer. The measurements were carried out by keeping the sample temperature at 9 K in a closed cycle helium cryostat.

## 3. Results and Discussion

### 3.1. Structural Characterisation of 1.5Eu^3+^: NaGdF_4_ NPs

Stable aqueous suspensions of NaGdF_4_ nanoparticles, undoped and doped with 1.5Eu^3+^, were successfully prepared at two reaction times (20 h and 24 h). Then, 1.5Eu^3+^: NaGdF_4_-20 and 1.5Eu^3+^: NaGdF_4_-24 nanoparticle suspensions were centrifuged and dried at 75 °C overnight to obtain the powders. Both powders were heat-treated at 450 °C for 5 h and characterised by X-ray diffraction in the range 10° < 2Ɵ < 70°. The XRD diffractograms, shown in Figure 1a, confirm the crystallisation of NaGdF_4_ in the hexagonal phase (JCPDS 28-1085) as the only appearing phase. Nevertheless, the definition of the peaks increases when increasing the reaction time from 20 to 24 h, suggesting bigger and better crystallised nanoparticles. The differences observed in the intensities of the peaks of XRD patterns are associated with a preferential orientation of the nanoparticles. This phenomenon can occur mainly due to the method of preparing the XRD samples. In the present work, the XRD samples were prepared by pressing the powder on a glass substrate with a small amount of Vaseline. Thus, the pressing procedure on the glass substrate could generate the preferential orientation of the nanoparticles. However, this procedure was chosen because the preparation of samples using other routes, such as putting the powder directly on the sample holder, could lead to the contamination of the XRD equipment [45]. 

The particle size, calculated using the Scherrer equation, increased from 30 nm to 70 nm for a reaction time of 20 to 24 h, respectively. He, F. et al. [46] described the preparation of hexagonal NaGdF_4_ nanoparticles via hydrothermal synthesis where the crystal size increased from 170 nm to 1 µm when the reaction time increased from 2 h to 24 h. The growth of NaGdF_4_ nanoparticles during hydrothermal synthesis is associated with a nucleation process followed by a crystal growth. Thus, the nucleation rate determines the growing crystal rate, and both processes are in competition [47].

To avoid a scattering process associated with large nanoparticle size or their agglomerations, a reaction time of 20 h was selected for the rest of the studies, to ensure particle sizes below 30 nm.

Furthermore, 1.5Eu^3+^: NaGdF_4_-20 nanoparticle suspension was sintered and heat-treated at 450 °C or 550 °C for 5 h. Figure 1b shows the XRD pattern of 1.5Eu^3+^: NaGdF_4_-20 NPs heat-treated at 450 °C and 550 °C for 5 h. In the case of 1.5Eu^3+^: NaGdF_4_-20 nanoparticles heat-treated at 550 °C, a partial transformation of hexagonal to cubic (JCPDS 27-0697) and orthorhombic (JCPDS 33-1007) phases is observed, associated with the presence of peaks at 2θ = 31° and 56°, while only the hexagonal phase is detected at 450 °C. For this reason, a sintering condition of 450 °C for 5 h was chosen to ensure the presence of the hexagonal phase as unique.

Figure 1c shows the W-H plot corresponding to the 1.5Eu^3+^: NaGdF_4_-20 and 1.5Eu^3+^: NaGdF_4_-24, both heat-treated at 450 °C for 5 h. The strain of the crystallites was calculated from the slope of each linear regression, and the strain decreases from 0.0986 to 0.0561 for 1.5Eu^3+^: NaGdF_4_-20 to 1.5Eu^3+^: NaGdF_4_-24, respectively. Moreover, the particle size, calculated from the y-intercept, increases from 16 nm for the sample 1.5Eu^3+^: NaGdF_4_-20 to 60 nm for 1.5Eu^3+^: NaGdF_4_-24. The fact that the crystal size calculated using the W-H plot for Eu^3+^: NaGdF_4_-20 is around half that calculated using the Scherrer equation likely indicates that a great part of the width of the XRD peaks corresponds to the intrinsic strain of the nanoparticles. By contrast, nanoparticles obtained with a high reaction time (1.5Eu^3+^: NaGdF_4_-24) present lower differences between the crystallite size calculated using the W-H and Scherrer equations, evidencing smaller strain contributions, probably due to nanoparticles with a bigger size and higher molecular order.

Figure 2 shows the HR-TEM images corresponding to 1.5Eu^3+^: NaGdF_4_-20 nanoparticles heat-treated at 450 °C for 5 h. Figure 2a shows nanoparticles with irregular shapes and sizes around 30 nm, around double of that calculated with the W-H plot, which is associated with the tendency of nanoparticles to agglomerate during synthesis and further centrifugation to obtain powders. The hexagonal shape is not clearly observed, probably due to the low amount of phase present in the nanoparticles. Wu, Y. et al. [48] reported the hydrothermal preparation of hexagonal NaGdF_4_ NPs using the same temperature and reaction times, observing that when NaF is used as a fluoride precursor, and for pH = 7, nanoparticles tend to have irregular shapes, as shown in this work. The size distribution is shown in Figure 2b, and a broad peak is observed with NP sizes between 20 nm and 42 nm with an average size of 34 nm. Figure 2c shows an amplified image of Figure 2a, revealing the existence of pores of around 2.5–3 nm inside the NP. The presence of pores can be attributed to a rapid formation of the nanoparticles followed by a long time for growing during which pores can be formed, as mentioned before [49]. The corresponding fast Fourier transform (FFT) pattern from the marked area in Figure 2d displays an interplanar distance of nearly 0.36 nm associated with the (001) plane of the NaGdF_4_ hexagonal phase, confirming the XRD results.

The structural characterisation of the 1.5Eu^3+^: NaGdF_4_ nanoparticles reveals that, even though 1.5Eu^3+^: NaGdF_4_-24 nanoparticles show a higher crystallisation degree, those synthesised for 20 h are more suitable for being incorporated into silica sol due to their smaller particle size. Moreover, a heat treatment of 450 °C for 5 h was chosen for these nanoparticles to avoid undesired crystallisation together with the hexagonal NaGdF_4_. For these reasons, 20 h was selected as the best reaction time to prepare NaGdF_4_ nanoparticle suspensions to carry out the further preparation of the OxGCs, and a heat treatment of 450 °C for 5 h as the best condition for sintering and performing the further characterisations.

### 3.2. Structural Characterisation of 80SiO_2_-20(1.5 Eu^3+^: NaGdF_4_) OxGCs Powders

OxGCs with the composition 80SiO_2_-20(1.5Eu^3+^-NaGdF_4_) were prepared through the incorporation of 1.5Eu^3+^: NaGdF_4_-20 nanoparticles in aqueous suspension in the silica sol precursors. For XRD characterisation, the 80SiO_2_-20(1.5Eu^3+^: NaGdF_4_) sol was dried and heat-treated at 450 °C for 5 h. Figure 3 shows the presence of hexagonal (JCPDS 28-1085) and orthorhombic (JCPDS 33-1007) phases. The partial phase transformation can be attributed to the change in pH during the preparation of the sol, from pH 7 of NP suspension to pH 2 of the final sol. Some authors reported that the nanoparticles with the formula NaLnF_4_ (Ln = Gd, Y, Li) are very sensitive to pH changes [46,50]. 

Figure 4a shows the FTIR spectra obtained for undoped and 1.5Eu^3+^-doped NaGdF_4_ nanoparticles heat-treated at 450 °C for 5 h (violet and blue line, respectively) and 80SiO_2_-20(1.5 Eu^3+^: NaGdF_4_) heat-treated at 450 °C for 5 h and dried at 75 °C overnight (red and green line, respectively) in the range of 4000 cm^−1^ to 400 cm^−1^. In all cases, no clear peaks were detected in the range of 3000 cm^−1^–4000 cm^−1^, indicating that the materials were free of water (-OH bonds should appear around 3000–3500 cm^−1^). Figure 4b shows the amplified spectra between 1500 cm^−1^ and 400 cm^−1^. In the 1.5Eu^3+^-doped NaGdF_4_ nanoparticles, at around 500 cm^−1^, a rising peak is identified associated with the tetrafluoride bond vibration [51] and confirming the formation of NaGdF_4_ nanoparticles. This peak is not completely observed due to the range limitation of the equipment. A small peak at 807 cm^−1^ and a broad double peak around 1114 cm^−1^ are also observed, associated with traces of Gd-O bonds, indicating the presence of oxygen defects in the nanoparticle lattice [52,53]. These defects could be due to the reactions of acetates with nitrate precursors, since these peaks appear for both doped and undoped NP samples. These traces can affect the luminescence properties [54]. On the other hand, in the spectrum corresponding to the OxGC dried at 75 °C (green plot), a sharp band is identified at about 1260 cm^−1^ together with a small band at 600 cm^−1^ associated with Si-CH_3_ groups from the silica precursor. The band at 1260 cm^−1^ remains after the heat treatment (red plot), indicating the presence of Si-CH_3_ groups. In both OxGCs spectra (green and red plots), a broad band between 1200 and 1000 cm^−1^ is assigned to Si-O-Si bonds, with overlapping bands such as the 1040 cm^−1^ and 1170 cm^−1^ peaks associated with the transversal optical (TO) and longitudinal optical (LO) asymmetric stretching modes of Si-O-(Si), respectively, and possible Gd-O bonds (at 1114 cm^−1^) overlapped with Si-O-Si vibrations, also identified in the 1.5Eu^3+^: NaGdF_4_ NPs (blue plot). On the other hand, a small broad band is observed around 930 cm^−1^, associated with Si-O(H) stretching vibration and/or non-bridging oxygen vibration. This band disappears after the heat treatment of OxGC, confirming the cross-linking of the SiO_2_ network. A further band around 800 cm^−1^ corresponds to the deformation vibration of silicate tetrahedrons (O-Si-O) and Gd-O bonds. The vibration band appearing at 472 cm^−1^ is assigned to deformation vibrations of silicate tetrahedrons, ν4 (O-Si-O), which becomes better defined after the heat treatment of OxGC (red plot). Finally, in both OxGCs spectra, the band around 400 cm^−1^ associated with crystalline tetrafluoride, also identified in the NP spectrum (blue line), is difficult to assign due to the proximity of the silicate tetrahedrons band and the limitation of the equipment [13].

These results confirm the successful incorporation of the 1.5Eu^3+^: NaGdF_4_ nanoparticles into the silica sol as their presence was evidenced both in the XRD pattern and FTIR spectra. However, the partial crystal transformation that takes place during the formation of the sol, from hexagonal to orthorhombic, should affect the optical properties of the OxGCs due to a reduction in the crystal symmetry. Nevertheless, molecular characteristics of the nanoparticles, such as the oxygen defect revealed in the FTIR spectra, suggest that the hexagonal nanoparticles that remain keep their properties throughout the sol formation.

### 3.3. Luminescence Properties of 1.5Eu^3+^: NaGdF_4_ NPs

Figure 5a shows the excitation spectrum of the heat-treated 1.5 Eu^3+^: NaGdF_4_ NPs recorded by monitoring the ^5^D_0_→^7^F_2_ emission of Eu^3+^ at 615nm. The spectrum shows a broad band centred around 245 nm attributed to the Eu^3+^-O_2_-charge transfer band (CTB). The CTB reflects the accommodation of oxygen into a lattice of the NaGdF_4_ NPs, which has already been reported for fluoride nanoparticles prepared via wet chemistry [38], and it is in concordance with the FTIR spectra shown in Section 3.2, where oxygen bonds have been identified. In addition to the CTB, the spectrum shows a peak at 272 nm attributed to the ^8^S_7/2_→^6^I_J_ transition of Gd^3+^ and the ^7^F_0_→ ^5^L_6_ peak of Eu^3+^ at 395 nm. The presence of the Gd^3+^ peak in the excitation spectrum obtained by monitoring the Eu^3+^ luminescence confirms the Gd-Eu energy transfer. The other two small peaks at 423 and 501 nm are from the xenon lamp used for excitation.

The emission spectra of the 1.5 Eu^3+^: NaGdF_4_ NPs, shown in Figure 5b, were obtained under excitation at 245 nm (CTB) and 272 nm (^8^S_7/2_→^6^I_J_) (Gd^3+^), respectively. The spectra show the Gd^3+^ emission (^6^P_0_→^8^S_7/2_) around 311 nm together with the Eu^3+^ emissions. The Gd^3+^ emission appears superimposed to a weak broad band centred at around 360 nm, likely associated with the 4f^6^5d-4f^7^ transition of Eu^2+^. For both excitations, the main emissions correspond to the ^5^D_0_→^7^F_J_ transitions of Eu^3+^. A small contribution of the ^5^D_1,2,3_→^7^F_J_ transition is also observed. The observed wavelength emissions, shown in Table 1, are similar to those published in the literature for other fluoride phases, such as GdF_3_ or NaLaF_4_. Moreover, the emission spectrum obtained after excitation through the charge transfer band (λ_exc_ = 245 nm) is much higher in intensity than that obtained after Gd^3+^ excitation at 272 nm, in accordance with the most efficient energy transfer.

The most intense peak of the spectra corresponds to the electric dipole ^5^D_0_→^7^F_2_ transition, indicating that Eu^3+^ ions are in a non-centrosymmetric site; the intensity of this hypersensitive transition is strongly affected by the local field [55]. The asymmetry ratio, R, defined as the ratio of the ^5^D_0_→^7^F_2_ and ^5^D_0_→^7^F_1_ emission intensities, was calculated to describe this behaviour. The lower this value, the closer the local symmetry is to the one with an inversion centre [55]. The obtained value of 1.46 confirms the non-centrosymmetric site, which can be attributed to the substitution of a F^−^ ion by an O^2−^ ion in the coordination environment of an Eu^3+^ which lowers the site symmetry of the Eu^3+^ in the oxygen-free NaGdF_4_ crystal [54]. 

In addition, the quantum yield calculated under excitation at λ_exc_ = 270 nm for the 1.5 Eu^3+^: NaGdF_4_ NPs gives a value of 43.1%. The quantum yield of Eu^3+^-doped fluorides with Gd^3+^ ions has been calculated by many authors, most of them reporting a theoretical value of 200% due to the energy transfer between Gd^3+^ and Eu^3+^ with exciting Gd^3+^ [56,57,58]. The low emission efficiency after Gd^3+^ excitation, evidenced through the quantum yield value, could be due to the presence of oxygen impurities in the NaGdF_4_ nanoparticles, as it is known that even a small amount of oxygen in the vicinity of Eu^3+^ ions can lead to a reduction in the photon emission [9].

The experimental decay curve from the ^5^D_0_ level was obtained by exciting at 245 nm through the CTB and collecting the luminescence of the ^5^D_0_→^7^F_2_ transition at 615 nm. The decay curve, displayed in Figure 6, shows a rise time of 0.44 ms, due to the population of the ^5^D_0_ level from higher energy levels, followed by a single exponential decay with a lifetime of 9.6 ms. This lifetime value is longer than some of those reported for hexagonal nanoparticles with the composition Eu^3+^: NaYF_4_ or Eu^3+^: NaGdF_4_ [59,60,61] and is similar to that found by Karbowiak et al. for Eu^3+^: NaGdF_4_ nanocrystals [62].

### 3.4. Luminescence Properties of 80SiO_2_-20(1.5Eu^3+^NaGdF_4_) OxGCs

The luminescent properties of the OxGCs powders with the composition 80SiO_2_-20(1.5Eu^3+^-20NaGdF_4_) heat-treated at 450 °C for 5 h are presented in Figure 7a, which shows the emission spectrum recorded under excitation at 245 nm. The spectral features are similar to those obtained for 1.5Eu^3+^: NaGdF_4_ NPs, with the peaks revealing the transition ^6^P_0_→^8^S_7/2_ of Gd^3+^ and the ^5^D_0_→^7^F_J_ transition from the Eu^3+^ ion. In addition to the nanoparticles study, the most intense peak was found to be corresponding to the ^5^D_0_→^7^F_2_ transition, in agreement with a non-centrosymmetric site for the Eu^3+^ ion. In this case, the calculated R_-_value is also 1.46, which is similar to that of the NPs. Furthermore, the wavelength of the emissions is the same as that detected for the nanoparticles reported in the previous section.

Figure 7b shows the luminescence decay curve from the ^5^D_0_ level recorded while collecting the emission at 615 nm and exciting the CTB at 245 nm. The wavelength emissions are listed in Table 1. The lifetime value obtained was 7.9 ms, which was lower than the one obtained for the nanoparticles (9.6 ms). The decrease in the lifetime value could be due to the presence of the remaining organics in the silica matrix from the sol-gel precursor, as shown in the FTIR spectra of Section 3.2, where a peak of CH_3_ from the MTES was identified. Moreover, the rising time was 0.4 ms, which was similar to that from the NPs.

**Table 1 nanomaterials-13-00940-t001:** Emission wavelengths corresponding to the observed transitions of Gd^3+^ and Eu^3+^ in NPs and OxGCs.

Observed Transitions	Wavelength (Figure 5b and Figure 7a)	NaGdF_4_ [63]	NaLaF_4_ [64,65]	EuF_3_ [66]	GdF_3_ [67]
** ^6^ ** **P_0_** **→** ** ^8^ ** **S_7/2_**	311 nm				
** ^5^ ** **D_3_** **→** ** ^7^ ** **F_1_**	415.5 nm				
** ^5^ ** **D_3_** **→** ** ^7^ ** **F_2_**	429 nm				
** ^5^ ** **D_3_** **→** ** ^7^ ** **F_3_**	434.5 nm				
** ^5^ ** **D_3_** **→** ** ^7^ ** **F_4_**	468 nm				
** ^5^ ** **D_2_** **→** ** ^7^ ** **F_2_**	478.5 nm				
** ^5^ ** **D_2_** **→** ** ^7^ ** **F_3_**	505.5				
** ^5^ ** **D_1_** **→** ** ^7^ ** **F_0_**	525 nm				
** ^5^ ** **D_1_** **→** ** ^7^ ** **F_1_**	534.5 nm				
** ^5^ ** **D_1_** **→** ** ^7^ ** **F_2_**	554 nm				
** ^5^ ** **D_0_** **→** ** ^7^ ** **F_1_**	583 nm, 591 nm	592 mm	592 mm	590 nm	591 nm
** ^5^ ** **D_0_** **→** ** ^7^ ** **F_2_**	614.5 nm	613 nm	615 nm	612/617 nm	615 nm
** ^5^ ** **D_0_** **→** ** ^7^ ** **F_3_**	649 nm	653 nm	650 nm		652 nm
** ^5^ ** **D_0_** **→** ** ^7^ ** **F_4_**	689 nm, 695.5 nm	695 nm	695 nm		

As mentioned before, the optical properties of trivalent Eu are highly sensitive to the local environment. Since the ^5^D_0_ state is non-degenerative under any symmetry, the structure of the ^5^D_0_→^7^F_J_ emissions is only determined by the splitting of the terminal levels caused by the local crystal field. Moreover, as the ^7^F_0_ level is also non-degenerative, site-selective excitation within the ^7^F_0_→^5^D_0_ absorption band can be performed using fluorescence line narrowing (FLN) spectroscopy, useful for distinguishing between different local environments around the Eu^3+^.

Time-resolved fluorescence line-narrowed (TRFLN) spectra of the ^5^D_0_→^7^F_0-2_ transitions of 80SiO_2_-20(1.5Eu^3+^: NaGdF_4_) powders heat-treated at 450 °C for 5 h were obtained at 9 K using different resonant excitation wavelengths throughout the ^7^F_0_→^5^D_0_ transition with a time delay of 10 μs. Depending on the excitation wavelength, the emission spectra present different characteristics, mainly related to the relative intensity and the level splitting of the transitions.

Figure 8 shows the low temperature (9 K) ^5^D_0_→^7^F_0-2_ emission spectra, obtained under excitation at 578.6 nm and 579 nm, both with a time delay of 10 µs after the laser pulse. These excitation wavelengths correspond to the highest emission intensities, with the spectrum obtained at 578.6 nm being more intense. Under excitation at 578.6 nm (Figure 8a), the ^5^D_0_→^7^F_1_ transition displays three stark components, which is in agreement with an Eu^3+^ occupying a site with orthorhombic symmetry, point group C_2v_, or lower [68]. The spectrum obtained under 579 nm excitation (Figure 8b) shows two components for the ^5^D_0_→^7^F_1_ transition. The presence of two components for the splitting of the ^7^F_1_ level and three for the ^7^F_2_ level is compatible with trigonal site symmetry (C_3,_ C_3v_) for Eu^3+^ ions. It is well known that the expected space group for β-NaGdF_4_ is P$\bar{6}$ where Gd^3+^ occupy point symmetry C_3h_ sites [38]. In this point symmetry, the ^5^D_0_→^7^F_0_ transition is forbidden and the number of lines for the ^5^D_0_→^7^F_1,2_ transitions are two and one, respectively [68]. However, the spectrum in Figure 8b shows the presence of the resonant ^5^D_0_→^7^F_0_ line as well as two and three components for the ^5^D_0_→^7^F_1,2_ emissions, respectively. This implies that a breaking symmetry occurs produced by lattice distortions which may reduce the original C_3h_ symmetry to C_3_ according to the branching rules of the 32 point groups [69]. The presence of two distinguishable sites for Eu^3+^ ions is, therefore, compatible with the observed hexagonal and orthorhombic phases detected in the XRD patterns of the OxGCs powders.

## 4. Conclusions

Stable 1.5Eu^3+^: NaGdF_4_ nanoparticle suspensions were successfully prepared via hydrothermal synthesis.

Nanoparticle synthesis is described as a process consisting of a first nucleation step, followed by a competition with the crystal growth.

XRD confirmed that hexagonal NaGdF_4_ is the only present phase after sintering at 450 °C for 5 h.

FTIR spectra revealed that 1.5Eu^3+^: NaGdF_4_ nanoparticles synthesised for 20 h present oxygen defects, probably forming Gd-O bonds.

In addition, nanoparticles present pores, likely due to the long synthesis process during which the crystal growth needs a much longer time than the nucleation step.

Although it is still possible to fit synthesis parameters to obtain better crystallised and defect-free hexagonal NaGdF_4_ nanoparticles, these results are promising, opening a new route to prepare NaGdF_4_ nanoparticle aqueous suspensions without any addition of organic dispersant.

It was confirmed that the acid media in which the OxGCs were prepared affects the stability of NaGdF_4,_ generating a partial crystal transformation from hexagonal to orthorhombic.

Emission and excitation spectra together with the lifetimes of the ^5^D_0_ state were measured for NPs and OxGCs with 1.5% Eu^3+^. The excitation spectra showed the Eu^3+^-O^2−^ charge transfer band in accordance with the presence of oxygen in the lattice of the NPs. The emission spectra obtained by exciting the charge transfer band show similar features in both NPs and OxGCs, corresponding the higher emission intensity to the ^5^D_0_→^7^F_2_ transition which indicates a non-centrosymmetric site for Eu^3+^. The reduction in the lifetime value of the ^5^D_0_ level from 9.6 ms in the NPs to 7.9 ms in OxGCs could be attributed to the presence of the remaining organics in the silica matrix. TRFLN spectra obtained under selective excitation in the ^7^F_0_→^5^D_0_ absorption band at 9 K allow us to identify the existence of two distinguishable sites for Eu^3+^ ions in the OxGCs powders with C_2v_ and C_3_ symmetries, which are compatible with the observed hexagonal and orthorhombic phases detected in the XRD diffraction patterns. 

## Figures and Tables

**Figure 1 nanomaterials-13-00940-f001:**
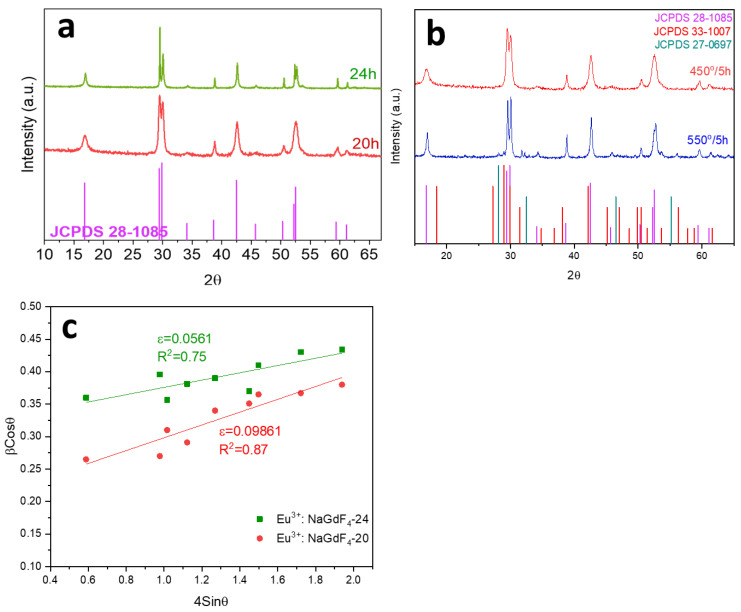
XRD patterns of (**a**) 1.5Eu^3+^: NaGdF_4_ NPs synthesised for 20 h and 24 h at 180 °C in the autoclave; (**b**) 1.5Eu^3+^: NaGdF_4_ NPs synthesised for 20 h at 180 °C and heat-treated at 450 °C and 550 °C for 5 h. (**c**) Williamson–Hall plot from the XRD of 1.5Eu^3+^: NaGdF_4_ NPs synthesised for 20 h and 24 h.

**Figure 2 nanomaterials-13-00940-f002:**
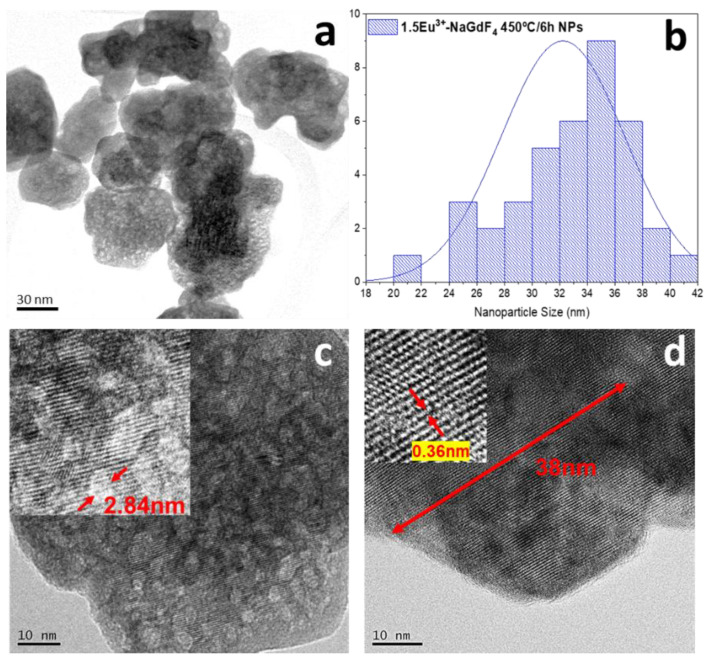
HR TEM images of (**a**) 1.5Eu^3+^: NaGdF_4_ NPs heat-treated at 450 °C for 5 h; (**b**) the corresponding nanoparticle size distribution taken from the image on “a”; (**c**) amplification of the area selected in “a” with the porous size measurement; (**d**) amplification of the area selected in “a” with the lattice distance measurement.

**Figure 3 nanomaterials-13-00940-f003:**
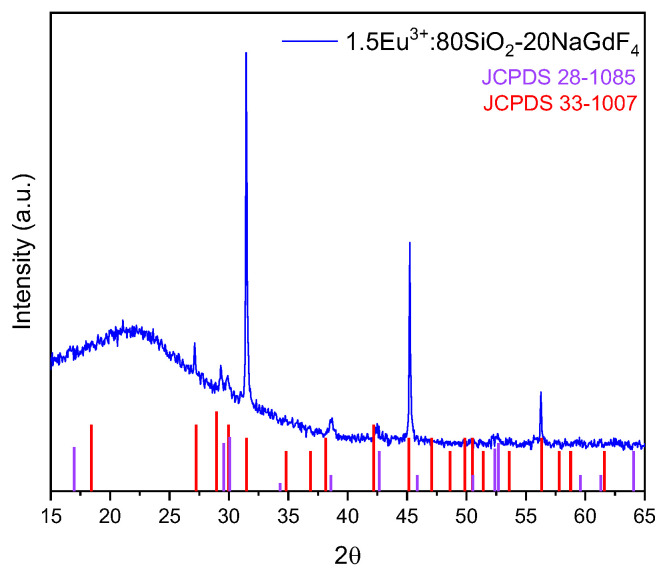
XRD pattern of OxGCs with the composition 80SiO_2_-20(1.5Eu^3+^: NaGdF_4_) heat-treated at 450 °C for 5 h.

**Figure 4 nanomaterials-13-00940-f004:**
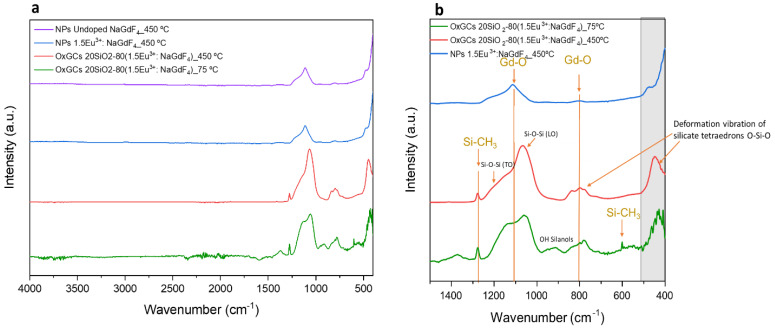
(**a**) FTIR spectra from 4000 cm^−1^ to 450 cm^−1^ of undoped NaGdF_4_ NPs, 1.5Eu^3+^: NaGdF_4_ NPs (violet and blue line, respectively) and OxGCs with the composition 80SiO_2_-20(1.5Eu^3+^: NaGdF_4_) dried overnight at 75 °C and heat-treated at 450 °C for 5 h (red and green line, respectively) and (**b**) FTIR spectra recorded from 1500 to 400 cm^−1^ of 1.5Eu^3+^: NaGdF_4_ NPs and 80SiO_2_-20(1.5Eu^3+^: NaGdF_4_) OxGCs with the composition (blue, red and green line, respectively).

**Figure 5 nanomaterials-13-00940-f005:**
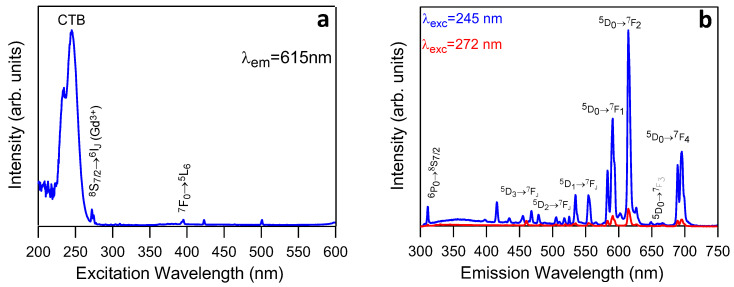
(**a**) Excitation spectrum of Eu^3+^ and Gd^3+^ ions from 1.5Eu^3+^: NaGdF_4_ NPs heat-treated at 450 °C for 5 h, collecting the luminescence at 615 nm. (**b**) Emission spectra of Eu^3+^ and Gd^3+^ ions from 1.5Eu^3+^: NaGdF_4_ NPs heat-treated at 450 °C for 5 h, excited at 245 nm (blue line) and 272 nm (red line).

**Figure 6 nanomaterials-13-00940-f006:**
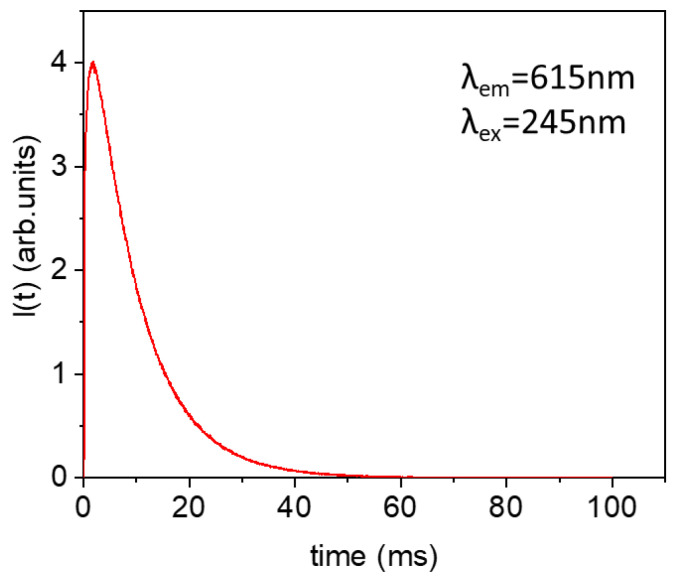
Experimental decay curve from ^5^D_0_ level of Eu^3+^ in 1.5Eu^3+^: NaGdF_4_ NPs heat-treated at 450 °C for 5 h, taken under excitation at 245 nm and collecting the luminescence at 615 nm.

**Figure 7 nanomaterials-13-00940-f007:**
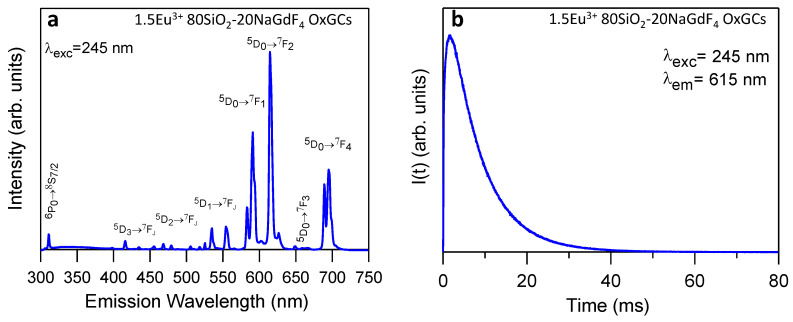
(**a**) Emission spectrum of Eu^3+^ and Gd^3+^ ions from OxGCs powders with the composition 80SiO_2_-20(1.5Eu^3+^: NaGdF_4_) under excitation at 245 nm and (**b**) experimental decay curve from the ^5^D_0_ level obtained by collecting the luminescence at 615 nm.

**Figure 8 nanomaterials-13-00940-f008:**
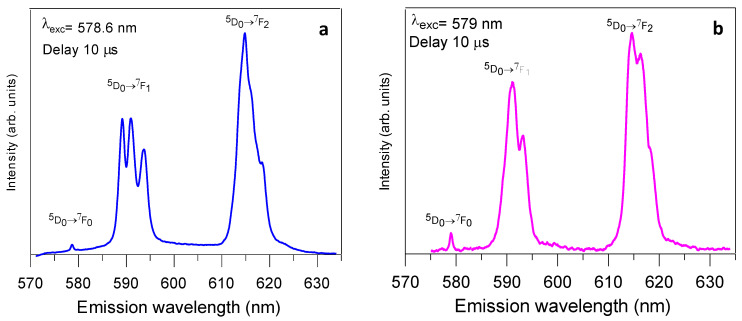
Time-resolved fluorescence line-narrowed emission spectra of the ^5^D_0_→^7^F_0,1,2_ transitions of Eu^3+^ ions measured at 9 K at a time delay of 10 µs after the laser pulse under an excitation at (**a**) 578.6 nm and (**b**) 579 nm, respectively, for the 80SiO_2_-20(1.5Eu^3+^: NaGdF_4_) powders heat-treated at 450 °C for 5 h.

## Data Availability

Data are available upon request.

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
