# Peer review of "Novel Sol-Gel Route to Prepare Eu3+-Doped 80SiO2-20NaGdF4 Oxyfluoride Glass-Ceramic for Photonic Device Applications"

_nanomaterials, 2023, doi:10.3390/nano13050940_

Round 1

Reviewer 1 Report

The manuscript is about Eu3+ doped oxyfluoride glass-ceramic.

In general, the structure of the manuscript is well organized, the methods are well described, and the procedures are well conducted. The results sound and the conclusions are based on the experimental data.

Comments on the manuscript:

l.93: Correction needed, °C and not ºC. Correction needed all along the text.

l.126: Was the washing step not necessary for this route?

l.144-145: Why 450-550 °C and 4-5 h as heat-treatment temperatures and holding times were used?

Fig.1(c): Give the R² for both plots.

Fig.3(b): How many images were taken to calculate the NP size distribution?

l.337: Care with sub and superscripts in chemical formula, ions. Revise the text for that.

References are old; from 53 references, only 17 are newer than 2016.

Author Response

Reviewer 1

  • 93: Correction needed, °C and not ºC. Correction needed all along the text.

According to the reviewer’s suggestion, all the “ºC” have been replaced by “°C”.

  • 126: Was the washing step not necessary for this route?

The washing step was necessary after the centrifugation of nanoparticles using deionized water and repeating the process three times. This procedure has been incorporated in the experimental part, Section 2.3. The following sentence:

“1.5Eu3+: NaGdF4-20 and 1.5Eu3+: NaGdF4-24 suspensions were centrifuged at 6000 rpm for 5 min, and the resulted powders rinsed with deionized water; the rising process was repeated three times.”

  • 144-145: Why 450-550 °C and 4-5 h as heat-treatment temperatures and holding times were used?

After the incorporation of the nanoparticles into a silica sol-gel, it is necessary to consolidate/ densify the silica matrix, made with a heat treatment at 450-500 ºC, usual in sol-gel materials. In our case, the NPs have been characterized at the same consolidation temperature as the silica matrix to analyze if the nanoparticles suffer any modification and if they are suitable for incorporation into the silica matrix.

  • 1(c): Give the R² for both plots.

Following the suggestion of the reviewer, the authors have incorporated the R2 values in Figure 1c, Section 3.1.

  • 3(b): How many images were taken to calculate the NP size distribution?

The authors used 10 images for obtaining an adequate NP size distribution (Figure 2, revised version). This aspect has been included in the experimental part, Section 2.3. The text in the revised version is:

“High Resolution Transmission Electron Microscopy (HRTEM) was used to characterize 1.5Eu3+: NaGdF4 NPs heat-treated at 450 °C for 5 h. The powder was re-dispersed in ethanol followed by dropping on a carbon-coated copper grid (Lacey Carbon, LC-200-Cu 25/pk). HRTEM images were taken from HRTEM-JEO 2100 microscope and the particle size distribution was determined using the ImageJ ® software and using a maximum of 10 images. The lattice parameters of NaGdF4 were determined using the Image-J source”

  • 337: Care with sub and superscripts in chemical formula, ions. Revise the text for that.

The authors have revised all the subs and superscripts.

  • References are old; from 53 references, only 17 are newer than 2016.

The authors have revised the references including some updated references. However, we have not removed any reference because we consider all relevant for the work.  

Reviewer 2 Report

1)The description is not concise, especially in the introduction.

2)The title is inappropriate. What does “highly efficient” mean? Is it luminescence efficiency of Eu3+ doped in 80SiO2-20NaGdF4 oxyfluoride glass-ceramic prepared by the pre-crystallized sol-gel route? But no reasonable supporting evidence is found for the text.

3)Why didn't the author provide the HR TEM images of 1.5Eu3+: NaGdF4 NPs in GCs? The purpose to study the properties of 1.5Eu3+: NaGdF4 NPs is unclear. The improvement of Eu3+ in 80SiO2-20NaGdF4 oxyfluoride glass-ceramic is hard to know, due to the lack of direct comparison between NPs and oxyfluoride glass-ceramic samples.

Author Response

Reviewer 2

  • The description is not concise, especially in the introduction.

According to the reviewer’s suggestions, the introduction has been modified and completed in the revised version.

  • The title is inappropriate. What does “highly efficient” mean? Is it luminescence efficiency of Eu3+ doped in 80SiO2-20NaGdF4 oxyfluoride glass-ceramic prepared by the pre-crystallized sol-gel route? But no reasonable supporting evidence is found for the text.

The title has been modified by “Novel sol-gel route to prepare Eu3+ doped 80SiO2-20NaGdF4 oxyfluoride glass-ceramic for future photonic devices applications”.

  • Why didn't the author provide the HR TEM images of 1.5Eu3+: NaGdF4 NPs in GCs? The purpose of studying the properties of 1.5Eu3+: NaGdF4 NPs is unclear. The improvement of Eu3+ in 80SiO2-20NaGdF4 oxyfluoride glass-ceramic is hard to know, due to the lack of direct comparison between NPs and oxyfluoride glass-ceramic samples.

The objective of the work is the preparation of OxGCs by sol-gel with high active phase, 80SiO2-20NaGdF4. This composition has not been reported in the literature using this alternative and innovative “pre-crystallized route”. So, the authors want to report that it is possible to prepare aqueous NaGdF4 suspensions to further incorporate them into a silica sol-gel sol. This process is not easy and requires the stabilization of the nanoparticles and then designing a process to incorporate them. Therefore, the study of nanoparticles is very important to understand if the NPs are stable. Thus, the nanoparticles have been studied by FTIR, DRX and HRTEM, and then the OxGCs have been also studied by FTIR and XRD and compared to NPs.  

In relation to the optical properties, the NPs and the OxGCs have been compared through the emission spectra and lifetime value, which has decreased from the NPs to the OxGCs.

The future work is the deposition of the OxGCs sol on glass-slide because it would be interesting the preparation of this composition as coating, but it is necessary to study the process step by step.

Reviewer 3 Report

Dear Authors,

The results presented in the article can be interesting and used in the development technologies of optical materials. I think there are valuable studies related to the synthesis method of NaGdF4 nanoparticles without the need for organics stabilizers.

Comments

1.      Section 2.3. information about the sample heating process should be supplemented. Were the samples heated in an oven (vacuum?) in an air atmosphere?

2.      I would recommend combining figure 2 and figure 5 into one figure. It will be easier to compare the FTIR spectra of all samples. The description of Figure 5 and the labels of the curves should be described in the same way.

3.      Corrections in lines 338-345  Eu3+ to Eu3+

Author Response

Reviewer 3

  • Section 2.3. Information about the sample heating process should be supplemented. Were the samples heated in an oven (vacuum?) in an air atmosphere?

The authors have included the information in the experimental part.

  • I would recommend combining figure 2 and figure 5 into one figure. It will be easier to compare the FTIR spectra of all samples. The description of Figure 5 and the labels of the curves should be described in the same way.

The authors agree with the suggestion of the reviewer and Figure 2 and Figure 5 have been combined in Figure 4 (revised version) and, the text in Section 3.2 has been modified, together with the caption.

  • Corrections in lines 338-345 Eu3+ to Eu3+

All the sub and superscript have been corrected.

Reviewer 4 Report

The authors presented the synthesis of Eu3+ doped fluorides and silica coated fluorides. They studied the XRD structures and FTIR, measured the emission spectra and investigated the fine-structure spectra. However, I don't feel any new physical insights in the manuscript. The presentation of the manuscript is not good. In fact, there are many technical incorrectness as partially highlighted in the attached manuscript. The manuscript was not acceptable in its current form. I suggest the authors to polish their work before its next submission.

Author Response

Reviewer 4

  • From the whole manuscript, there was no comparison of the Eu3+ emission between NaGdF4 and OxGCs. How to reveal the phrase 'highly efficient'. Besides, according to the experimental details, the authors seemed to synthesize the SiO2-NaGdF4 composites, which could not represent the oxyfluoride glass-ceramic systems. Were them in core-shell construction, or fluoride nanoparticles intercalated into SiO2? The authors should perform more characterizations such as TEM.

The authors have compared the emission spectra and lifetimes of the nanoparticles and OxGCs in Section 3.3 ( Luminescence properties of 1.5Eu3+: NaGdF4 NPs) and 3.4. (Luminescence properties of 80SiO2-20(1.5Eu3+:NaGdF4) OxGCs).

In relation to the title in the first version of the article “Highly efficient…”, it has been changed to: “Novel sol-gel route to prepare Eu3+ doped 80SiO2-20NaGdF4 oxyfluoride glass-ceramic for future photonic devices applications”

  • How did oxygen defects exist in NaGdF4 host? Most of the reports revealed the oxygen-based organic groups attached to the surface of the nanoparticles.

The oxygen could be incorporated during the synthesis, as defects, into the fluoride lattices and, when this happens, the nanoparticles are doped with Eu3+. This observation has been also reported before for the NaGdF4 nanoparticles. (1) Mech et al, (1) showed a FTIR spectra, where peaks near 1100 cm-1 are observed, without identification.

  • It was reported that NaGdF4/NaYF4 existed in two forms: cubic and hexagonal. How was the orthorhombic phase originated? One strong diffraction peak was not marked in the red cross in Fig 4, the authors should point out what this peak belongs to.

Figure 4 (Figure 3 in revised version) was modified incorporating the patterns of the hexagonal and orthorhombic phases and confirming the crystallization of both phases and the identification of all the peaks.

Some authors have reported the presence of NaGdF4 as cubic or hexagonal, but other authors have also reported the crystallization as orthorhombic and tetragonal for MGdF4 crystals (M=Li, Na, K). (2) In fact, in our case, the authors used the pattern of orthorhombic KGdF4 crystals to identify the phase. The possibility of matching the KGdF4 patterns as NaGdF4, considering a that K and Na ions can occupy the same place in the crystals, has been published before. (3,4)

On the other hand, in the sol-gel process it’s well-known that non-stable phases effectively crystallize. For example, for TiO2, the most thermal-stable phase is rutile but for sol-gel materials the most stable phase is anatase. So, it does not look strange to observe this effect in these nanoparticles. (5)

  • The QY of NaGdF4: Eu under 270 nm excitation has been measured and calculated. What about OxGCs? 245nm excitation? Please provide these data for further proving the high efficient emission of OxGCs: Eu samples.

The measurements of the OxGCs samples were not possible, due to the scattering generated by the silica matrix. Thus, only the nanoparticles measurement was performed.

  • From Fig. 9, the line-narrowed emission spectra revealed three components for each emission peak under both 578.6 nm and 578.8 nm excitation, the only difference is the emission profile, which can be realized by peak-splitting processing. The authors should provide detailed explanation of which structure should the TRFLN spectra responds to.

In accordance with the referee’s comment, Fig. 9b (original version) has been modified to show the TRFLN spectrum obtained under 579 nm excitation where the site is best resolved. Furthermore, to clarify the results of TRFLN spectra, part of the last paragraph before the conclusions has been rewritten and a new reference has been added (ref. 66 revised version): “The spectrum obtained under 579 nm excitation (Figure 8b) shows two components for the 5D0®7F1 transition. The presence of two components for the splitting of the 7F1 level and three for the 7F2 level is compatible with trigonal site symmetry (C3, C3v) for Eu3+ ions.65 It is well known that the expected space group for b-NaGdF4 is P where Gd3+ occupy point symmetry C3h sites.36 In this point symmetry the 5D0®7F0 transition is forbidden and the number of lines for the 5D0®7F1,2 transitions are two and one respectively.65 However, the spectrum in Fig. 8b shows the presence of the resonant 5D0®7F0 line as well as two and three components for the 5D0®7F1,2 emissions respectively. This imply that a breaking symmetry occurs produced by lattice distortions which may descent the original C3h symmetry to C3 according to the branching rules of the 32 point groups.66 The presence of two distinguishable sites for Eu3+ ions is therefore compatible with the observed hexagonal and orthorhombic phases detected in the XRD patterns of the OxGCs powders.”

As a consequence the last sentence of the conclusions “TRFLN spectra obtained under selective excitation in the 7F0 ®5D0 absorption band at 9 K allow to identify the existence of two distinguishable hexagonal and orthorhombic sites for Eu3+ ions in the OxGCs powders, in agreement with the observed hexagonal and orthorhombic phases detected in the XRD diffraction patterns”  has been substituted by   “TRFLN spectra obtained under selective excitation in the 7F0 ®5D0 absorption band at 9 K allow to identify the existence of two distinguishable sites for Eu3+ ions in the OxGCs powders with C2v and C3 symmetries, which are compatible with the observed hexagonal and orthorhombic phases detected in the XRD diffraction patterns

Reviewer 5 Report

 The manuscript describes the synthesis and structural and optical characterization of a silica matrix containing Eu-doped Gd oxyfluoride. The work may be published after the following points have been improved.

Better specify in the introduction that sol-gel techniques have been specially developed long times ago for the preparation of optical ceramics: the first example is the PLZT ceramic in 1976 (see Snow and other references in a recent review paper in Ceramics, 2020, 3,312- 339)

 Several summary points are unclear.

Explain the role of autoclaving for the preparation of NaGdF4

Line 130 : 26 superscript

Line 135: add more details to make reproduction of the synthesis possible

Line 179: Thermogravimetry, grain-size (SEM) or bET surface area measurements are needed to determine the water/hydroxyl content and the physical characteristics of the final product. Indeed Scherrer’ formula (line 216) gives information on the correlation length (size of ‘regular’ crystalline zone) and not of the grain size relevant for the control of the sintering behaviour. This is also confirmed by Fig 3 of TEM which shows heterogeneous particles of about 100 nm (see below).

Fig. 1: explicit the W-H acronym in the caption.

Line 250: The authors should give the spectra up to 4000 cm-1 to control hydroxylation/hydration of the compound. Otherwise the IR spectrum has no interest because the characteristic modes of the compound are out of the spectral window studied. Furthermore, the attribution of the band around 1100 cm-1 is not convincing; no fundamental mode of Gd2O3 is expected in this domain, this could be also related to hydroxylation. Without a study between 450 and at least 100 cm-1 and above 2500 cm-1 the IR spectrum is of no interest.

Figure 3b is not in agreement with 3a which shows composite particles of ~100 nm; the values ​​of the 3b distribution correspond to the sizes of the crystalline domains and not of the particles. This is best discussed by showing part of Fig 3a with higher magnification

Figure 5 and lines 305-325: very fine bands can also result from traces of nitrates; this point needs to be discussed. The heat treatment must eliminate the organic residues, which in any case must be clearly visible at around 3000 cm-1. Showing the IR spectrum around 3000 cm-1 is essential.

What is the final object and for what application: powder, layer or solid matrix, in the latter case (we are talking about ceramics in the abstract), what is its optical transmission, its porosity, a photo, dimensions, etc.? This point needs to be clarified and should be announced in the abstract and the introduction.

Figure 6 and 8: add a table with the characteristic wavelengths of the different transitions and comparison with literature of similar compounds

Author Response

Reviewer 5

  • Better specify in the introduction that sol-gel techniques have been specially developed long times ago for the preparation of optical ceramics: the first example is the PLZT ceramic in 1976 (see Snow and other references in a recent review paper in Ceramics, 2020, 3,312- 339)

The authors have incorporated two references indicating the importance of the sol-gel throughout the history in the preparation of Optical Materials.

  • Explain the role of autoclaving for the preparation of NaGdF4

According to the reviewer suggestion, a phase has been incorporated in the introduction about the use of the autoclave:

“Although, different papers have reported the preparation of powders NaGdF4 nanoparticles by hydrothermal synthesis in an autoclave controlling the pressure and temperature, (3,6,7) only a few reported the use of this process to obtain aqueous nanoparticles suspensions. This procedure is adequate to control the morphology of the nanoparticles.”

  • Line 130: 26 superscripts

All the sub and superscript have been corrected.

  • Line 135: add more details to make reproduction of the synthesis possible

The authors have modified the experimental part incorporating more details. Thus, the following sentence was completed to clarify all the steps:

The solution was immersed in an ice bath for 2 min to stop the reactions. After that, the sol was stirred for 15 min at room temperature”.

  • Line 179: Thermogravimetry, grain-size (SEM) or BET surface area measurements are needed to determine the water/hydroxyl content and the physical characteristics of the final product. Indeed Scherrer’ formula (line 216) gives information on the correlation length (size of ‘regular’ crystalline zone) and not of the grain size relevant for the control of the sintering behavior. This is also confirmed by Fig 3 of TEM which shows heterogeneous particles of about 100 nm (see below).

The authors described the physical characteristics of the materials through FRIT and XRD and, in the case of the nanoparticles, also by HR TEM. To deeply study the crystallite characteristics of the nanoparticles, the W-H plot has been done, understanding that Scherrer equation gives a limited information, as the reviewers suggests.

  • 1: explicit the W-H acronym in the caption.

According to the reviewer’s suggestion, the caption has been changed as follows: Figure 1. XRD patterns of (a) 1.5Eu3+: NaGdF4 NPs synthesized for 20 h and 24 h at 180 °C in the autoclave, (b) 1.5Eu3+: NaGdF4 NPs synthesized for 20 h at 180 °C and heat-treated at 450 °C and 550 °C during 5 h. (c) William-Hall plot from the XRD of 1.5Eu3+: NaGdF4 NPs synthesized for 20 h and 24 h.

  • Line 250: The authors should give the spectra up to 4000 cm-1 to control hydroxylation/hydration of the compound. Otherwise, the IR spectrum has no interest because the characteristic modes of the compound are out of the spectral window studied. Furthermore, the attribution of the band around 1100 cm-1 is not convincing; no fundamental mode of Gd2O3 is expected in this domain, this could be also related to hydroxylation. Without a study between 450 and at least 100 cm-1 and above 2500 cm-1 the IR spectrum is of no interest.

Figures 2 and 5 (Original version) have been combined in Figure 4 as suggested by the reviewer 3. In the figure 4, the spectra have been plotted in the range of 4000 cm-1 -450 cm-1, there are not peaks and therefore evidence of presence de O-H group close to 3200 cm-1.

  • Figure 3b is not in agreement with 3a which shows composite particles of ~100 nm; the values ​​of the 3b distribution correspond to the sizes of the crystalline domains and not of the particles. This is best discussed by showing part of Fig 3a with higher magnification

Figure 3a shows the nanoparticles agglomerated, as it is indicated by the reviewer, but this is due to the sample preparation procedure. To analyze the nanoparticles by HRTEM, it was necessary to centrifuge the nanoparticles, and then, made an additional heat treatment. During all of this process, it is not possible to avoid the agglomeration.

  • Figure 5 and lines 305-325: very fine bands can also result from traces of nitrates; this point needs to be discussed. The heat treatment must eliminate the organic residues, which in any case must be clearly visible at around 3000 cm-1. Showing the IR spectrum around 3000 cm-1 is essential.

The authors have been modified the FTIR spectra, now new Figure 4.

  • What is the final object and for what application: powder, layer or solid matrix, in the latter case (we are talking about ceramics in the abstract), what is its optical transmission, its porosity, a photo, dimensions, etc.? This point needs to be clarified and should be announced in the abstract and the introduction.

The authors agree with the reviewer and the abstract and introduction have been modified. Thus, the following text has been incorporated in the introduction:

“The objective of this work was the preparation and characterization of optically active oxyfluoride glass-ceramics through the incorporation of stable aqueous Eu3+-doped NaGdF4 nanoparticles suspensions into a silica sol. In this work, powdered glass-ceramics were prepared and the suitability of this processing method to prepare transparent OxGCs was demonstrated. This is a promising method for preparing transparent coatings and overcomes the drawbacks of the traditional melting-quenching method, adequate to obtain bulk materials but not able for preparing coatings.”

  • Figure 6 and 8: add a table with the characteristic wavelengths of the different transitions and comparison with literature of similar compounds

According with the referee´s comment a new Table (Table 1, revised version) has been added.

Reviewer 6 Report

The manuscript requires revision. The XRD phase identification should be reconsidered. Please see my comments below.

P. 5, line 209. Please change the phrase “Stables aqueous suspensions…” to “Stable aqueous suspensions…”

In Fig. 1(a), the relative intensities of the two main peaks do not match those presented in the standard XRD card. Intensity of the peak at 2theta at about 17 degrees is much lower than that in the standard XRD card. Please give your comments.

Line 229. “In the case of 1.5Eu3+: NaGdF4-20 nano-particles heat-treated at 550 ºC, a partial transformation of hexagonal to cubic phase is observed, associated with the presence of peaks at 2θ=31º and 56º, while only the hexagonal phase is detected at 450 ºC.” Please show the standard XRD card of the cubic phase. Please be aware that the main reflection of the cubic phase is absent in your XRD pattern. Therefore, the cubic phase is not formed after heat-treatment at 550 ºC.

Fig. 2. There are no peaks at wavenumbers higher than 1500 cm-1. Therefore, it seems reasonable to present the spectra in the spectral range of 1500 cm-1 to 400 cm-1 and discuss them in detail.

Line 267. Please explain the meaning of the sentence “A lack of defined hexagonal shape is also observed, probably related to the low definition of the peaks in XRD.”

Line 297. “Figure 4 shows the presence of hexagonal and orthorhombic (JCPDS 33-1007) phases.” Please reconsider the phase identification. The strong reflection at about 2theta=45 degrees is not assigned to a certain phase. The reflections assigned by the authors to the hexagonal phase are very weak; there are traces of this phase. The relative intensities of the reflections do not match those on the standard XRD card.

Fig. 4. To prove the crystallization of the orthorhombic phase, please add its standard card to the figure. Please assign the strong reflection at about 2theta=45 degrees to a certain crystalline phase.

Line 340. Please correct the typo in the phrase “The CTB reflects the accommodation of oxygen into a lattice of the NaGdF4 NPs, which has already been reported for fluoride nanoparticles prepared by wet-chemistry 31…”

Caption of Fig. 9. Please indicate the material from which the spectrum was recorded.

In conclusions, it is reasonable to mention the discrepancy between the standard XRD card of hexagonal NaGdF4 crystals and XRD patterns of the NaGdF4 NPs.

Line 455. Please correct the phrase “Although there it’s still possible to fit synthesis parameters…’’

Line 459. Please correct the phrase “The preparation of the OxGCs from the NPs suspension showed to affect the NaGdF4 stability due to the acid media in which are prepared,…”

Author Response

Reviewer 6

The manuscript requires revision. The XRD phase identification should be reconsidered. Please see my comments below.

  • 5, line 209. Please change the phrase “Stables aqueous suspensions…” to “Stable aqueous suspensions…”

According to the reviewer suggestion, the phrase “Stables aqueous suspensions.” was changed to “Stable aqueous suspensions…” in Section 3.1.

  • In Fig. 1(a), the relative intensities of the two main peaks do not match those presented in the standard XRD card. The intensity of the peak at 2theta at about 17 degrees is much lower than that in the standard XRD card. Please give your comments.

The differences observed in the intensities of the peaks of XRD could be associated with a preferential orientation of the nanoparticles. This phenomenon can occur mainly due to the way of preparing the XRD samples. In the present work, the XRD samples were prepared by pressing the powder on a glass substrate with a small quantity of Vaseline. Thus, the pressing procedure on the glass substrate could generate the preferential orientation of the nanoparticles. However, this procedure was chosen because the preparation of samples using other routes (putting the powder directly on the sample holder), could lead the contamination of the XRD equipment. (8)

  • Line 229. “In the case of 1.5Eu3+: NaGdF4-20 nanoparticles heat-treated at 550 ºC, a partial transformation of hexagonal to cubic phase is observed, associated with the presence of peaks at 2θ=31º and 56º, while only the hexagonal phase is detected at 450 ºC.” Please show the standard XRD card of the cubic phase. Please be aware that the main reflection of the cubic phase is absent in your XRD pattern. Therefore, the cubic phase is not formed after heat-treatment at 550 ºC.

The corresponding patterns were included in all the DRX figures, to clarify this point.

  • 2. There are no peaks at wavenumbers higher than 1500 cm-1. Therefore, it seems reasonable to present the spectra in the spectral range of 1500 cm-1 to 400 cm-1 and discuss them in detail.

The FTIR figure has been not modified because other two reviewers recommended to include the complete range.

  • Line 267. Please explain the meaning of the sentence “A lack of defined hexagonal shape is also observed, probably related to the low definition of the peaks in XRD.”

The authors modified the sentence to clarify the idea. “The hexagonal shape is not clearly observed, probably associated with the low crystallinity of the phase present in the nanoparticles, which has been also revealed in the low definition of the peaks in the XRD”.

  • Line 297. “Figure 4 shows the presence of hexagonal and orthorhombic (JCPDS 33-1007) phases.” Please reconsider the phase identification. The strong reflection at about 2theta=45 degrees is not assigned to a certain phase. The reflections assigned by the authors to the hexagonal phase are very weak; there are traces of this phase. The relative intensities of the reflections do not match those on the standard XRD card.
  • 4. To prove the crystallization of the orthorhombic phase, please add its standard card to the figure. Please assign the strong reflection at about 2theta=45 degrees to a certain crystalline phase.
  •  

Figure 4 has been modified and the phase patterns have been incorporated (Figure 3 revised version). The lack of coincidence with the relative intensities has been explained in question 2. It is likely related to the method of sample preparation that generates preferential orientation.

  • Line 340. Please correct the typo in the phrase “The CTB reflects the accommodation of oxygen into a lattice of the NaGdF4 NPs, which has already been reported for fluoride nanoparticles prepared by wet-chemistry 31…”

The authors have corrected the phrase.  

Caption of Fig. 9. Please indicate the material from which the spectrum was recorded.

The caption of Figure 9 (Figure 8 in Revised version) has been modified.

Figure 8. Time-resolved fluorescence line-narrowed emission spectra of the 5D0→7F0,1,2 transitions of Eu3+ ions measured at 9 K at a time delay of 10 µs after the laser pulse under excitation at (a) 578.6 nm and (b) 579 nm, respectively, for the 80SiO2-20(1.5Eu3+: NaGdF4) powders heat-treated at 450 °C for 5 h.

  • In conclusion, it is reasonable to mention the discrepancy between the standard XRD card of hexagonal NaGdF4 crystals and XRD patterns of the NaGdF4 NPs.

All the cards have been incorporated into the XRD images to clarify this point.

  • Line 455. Please correct the phrase “Although there it’s still possible to fit synthesis parameters…’’

According to the reviewer’s suggestion, the phrase has been corrected. The sentences in the text (Revision version) is: “Although it is still possible to fit synthesis parameters…”

  • Line 459. Please correct the phrase “The preparation of the OxGCs from the NPs suspension showed to affect the NaGdF4 stability due to the acid media in which are prepared”

The sentence was corrected to: “It was confirmed that the acid media in which the OxGCs were prepared, affects the NaGdF4 stability, generating a partial crystal transformation from hexagonal to orthorhombic”

Round 2

Reviewer 2 Report

Authors have completed the corresponding modifications.

Author Response

the authors have checked all the things

Reviewer 5 Report

The text could be published in the state. But for the interest of authors and readers one point remains to be improved. The authors have fairly well improved the text according to the reviews except for the introduction, which they would have seen by referring to the old articles suggested: the main interest of sol-gel methods for the preparation of optical materials is the possibility of controlling the homogeneity at the quasi-molecular scale. This should be better explained in the introduction and it is a pity that the authors do not check the homogeneity of the distribution of Eu ions. Perhaps this is provided for in other works

Author Response

Dear Editor,

I am writing to submit the revised manuscript entitled “Novel sol-gel route to prepare Eu3+ doped 80SiO2-20NaGdF4 oxyfluoride glass-ceramic for photonic devices applications”. by María Eugenia Cruz, Thi Ngoc Lam Tram, Alessandro Chiassera, Alicia Durán, Joaquín Fernández, Rolindes Balda and, Yolanda Castro. We want to thank the Referees for their valuable comments. We have incorporated all the corrections and suggestions in the text, indicated with the “Track changes” tool of MS Word. We hope that the manuscript is now improved.

We response to the referee comment in red color

Reviewer 5

The text could be published in the state. But for the interest of authors and readers one point remains to be improved. The authors have fairly well improved the text according to the reviews except for the introduction, which they would have seen by referring to the old articles suggested: the main interest of sol-gel methods for the preparation of optical materials is the possibility of controlling the homogeneity at the quasi-molecular scale. This should be better explained in the introduction and it is a pity that the authors do not check the homogeneity of the distribution of Eu ions. Perhaps this is provided for in other works.

According to the reviewer’s suggestion, the introduction has been slightly modified for mentioning the characteristic of sol-gel that allows the control of the chemistry of the material in the molecular scale. In addition, important references in the field have been incorporated. The authors consider that a longer introduction of sol-gel process is not relevant in this work.

Regarding the homogeneity of the distribution of Eu ions in the glass-ceramic matrix, the authors consider that it is adequate, taking into account the optical results obtained. If the Eu ions were not well distributed, the luminescence response would be poor or very low, and in this case, it is not occurring.    

“Sol-Gel (SG) appeared as an alternative method, emphasized in the chemistry of the process, to avoid the drawbacks of MQ. It is a very favorable process through which highly homogeneous materials can be obtained at low temperatures (<500 °C). Furthermore, various material forms, e.g. bulks, powders, and coatings can be processed through the hydrolysis and polycondensation of metal alkoxide pre-cursors such as tetraethyl orthosilicate (TEOS) in a solvent, typically alcohol. Sol-gel permits control the structures in the molecular scale of the materials along all the process, allowing the preparation of highly homogeneous materials.”

Reviewer 6 Report

In my first report, I asked the authors to reconsider the phase identification presented in Fig. 3. I mentioned that the strong reflection at about 2theta=45 degrees is not assigned to a certain phase. The reflections assigned by the authors to the hexagonal phase are very weak. The relative intensities of the reflections do not match those on the standard XRD card. The authors answered that “The differences observed in the intensities of the peaks of XRD could be associated with a preferential orientation of the nanoparticles. This phenomenon can occur mainly due to the way of preparing the XRD samples. In the present work, the XRD samples were prepared by pressing the powder on a glass substrate with a small quantity of Vaseline. Thus, the pressing procedure on the glass substrate could generate the preferential orientation of the nanoparticles. However, this procedure was chosen because the preparation of samples using other routes (putting the powder directly on the sample holder), could lead the contamination of the XRD equipment.” I believe it is important to add this explanation to the text of the manuscript. Otherwise the readers will be puzzled.

The manuscript requires moderate English changes. I leave some comments below.

P. 3. Please correct typo in the word “preparation” (it is written as “reparation”).

P. 16. Please correct the phrase “On the other hand, in the OxGC spectrum dried at 75°C…”

P. 21. Please correct the sentence “Furthermore, the wavelength of the emissions is the same detected for nanoparticles reported in the previous section.”

P. 25. Please correct typo in the symbol “Gd-O”. It is written as “G-O bonds”.

Author Response

Dear Editor,

I am writing to submit the revised manuscript entitled “Novel sol-gel route to prepare Eu3+ doped 80SiO2-20NaGdF4 oxyfluoride glass-ceramic for photonic devices applications”. by María Eugenia Cruz, Thi Ngoc Lam Tram, Alessandro Chiassera, Alicia Durán, Joaquín Fernández, Rolindes Balda and, Yolanda Castro. We want to thank the Referees for their valuable comments. We have incorporated all the corrections and suggestions in the text, indicated with the “Track changes” tool of MS Word. We hope that the manuscript is now improved.

We response to the referee comment in red color

Reviewer 6

In my first report, I asked the authors to reconsider the phase identification presented in Fig. 3. I mentioned that the strong reflection at about 2theta=45 degrees is not assigned to a certain phase. The reflections assigned by the authors to the hexagonal phase are very weak. The relative intensities of the reflections do not match those on the standard XRD card. The authors answered that “The differences observed in the intensities of the peaks of XRD could be associated with a preferential orientation of the nanoparticles. This phenomenon can occur mainly due to the way of preparing the XRD samples. In the present work, the XRD samples were prepared by pressing the powder on a glass substrate with a small quantity of Vaseline. Thus, the pressing procedure on the glass substrate could generate the preferential orientation of the nanoparticles. However, this procedure was chosen because the preparation of samples using other routes (putting the powder directly on the sample holder), could lead the contamination of the XRD equipment.” I believe it is important to add this explanation to the text of the manuscript. Otherwise, the readers will be puzzled.

According to the reviewer’s suggestion, the text and the reference has been incorporated in the revised article (First paragraph, section 3.1).

Regarding the XRD, the authors have tried to identify all the peaks, considering the best JCPDS. in this case, we find that the orthorhombic phase (JCPDS 33-1007) is the best to explain all the reflections, especially the peak at about 2theta=45 degrees, together with the hexagonal. In the future, we will try to use other techniques to verify this. We would like to thank the reviewer for this appreciation that opens the door to further research.

The manuscript requires moderate English changes. I leave some comments below.

  1. 3. Please correct typo in the word “preparation” (it is written as “reparation”).
  2. 16. Please correct the phrase “On the other hand, in the OxGC spectrum dried at 75°C…”
  3. 21. Please correct the sentence “Furthermore, the wavelength of the emissions is the same detected for nanoparticles reported in the previous section.”
  4. 25. Please correct typo in the symbol “Gd-O”. It is written as “G-O bonds”.

According to the suggestions made by the referee, all the changes have been done in the text, identified with the “track changes” of MS Wo
